# BCC-Based Mg–Li Alloy with Nano-Precipitated MgZn₂ Phase Prepared by Multidirectional Cryogenic Rolling

Qing Ji, Xiaochun Ma, Ruizhi Wu *, Siyuan Jin, Jinghuai Zhang and Legan Hou

Key Laboratory of Superlight Materials & Surface Technology (Ministry of Education), Harbin Engineering University, Harbin 150001, China
* Correspondence: rzwu@hrbeu.edu.cn

**Abstract:** In this study, we deformed the single β phase Mg–Li alloy, Mg–16Li–4Zn–1Er (LZE1641), with conventional rolling (R) and multi-directional rolling (MDR), both at cryogenic temperature. Results showed that the nano-precipitation phase MgZn₂ appeared in the alloy after MDR, but this phenomenon was not present in the alloy after R. The finite element simulation result showed that the different deformation modes changed the stress distribution inside the alloy, which affected the microstructures and the motion law of the solute atoms. The high-density and dispersively distributed MgZn₂ particles with a size of about 35 nm were able to significantly inhibit the grain boundary migration. They further hindered the dislocation movement and consolidated the dislocation strengthening and fine-grain strengthening effects. Compared with the compressive strength after R (273 MPa), the alloy compressive strength was improved by 21% after MDR (331 MPa). After 100 °C compression, the MgZn₂ remained stable.

**Keywords:** bcc Mg–Li alloy; cryogenic; multi-directional rolling; nano–grains; dislocation; MgZn₂ phase

## 1. Introduction

Magnesium (Mg) alloys not only have the advantages of low density, high specific strength, and high specific stiffness, but also have excellent damping and electromagnetic shielding properties [1,2]. These properties enable Mg alloys to adapt well to the high demands of aerospace, military equipment, medical equipment, and 3C electronics. However, in the production process, Mg alloys have a strong basal texture and are difficult to deform due to the close-packed hexagonal (hcp) lattice structure of Mg [3–5], which weakens their advantages when competing with other lightweight alloys.

The addition of Li into Mg alloys gradually changes the hcp structure to body-centered cubic (bcc) structure. Depending on the Li content, the lattice type of magnesium alloys present three states: α phase (<5.7 wt.%), α + β phase (5.7–10.3 wt.%), and β phase (>10.3 wt.%) [6–8]. Moreover, the density of Li is 0.53 g/cm³, which can further reduce the density of the Mg alloy. To cope with the strict demands for lightweight alloys, 14 wt.% or more Li is added to the Mg alloy. However, the addition of Li introduces a new problem, namely that it is difficult to improve the strength of the alloy beyond 200 MPa. The increase of active slip systems in bcc lead to the easy deformation of the alloy and the easy activation of dislocations [9]. Therefore, improving the strength of single-β-phase Mg–Li alloys has become a particularly urgent focus.

To overcome such problems, research has mainly focused on plastic deformation [10,11]. Plastic deformation processes such as extrusion, rolling, and torsion are effective means to improve the strength of β-phase magnesium–lithium alloys. The bcc structure has high stacking fault energy, wide spreading dislocations, and low critical shear stress (CRSS). These characteristics make the dislocation prone to slip [12]. However, to counteract the stress-concentration state, dynamic recrystallization (DRX) prematurely enters the stage of grain

growth after nucleation, which causes dislocations to be prematurely eliminated during the accumulation process. Thus, the effect of work-hardening and fine-grain strengthening cannot continue to proliferate with the amount of deformation.

Changing parameters such as temperature and deformation can effectively improve the strength of β-phase Mg–Li alloys by regulating DRX and dislocations [13–16]. Furthermore, solid solution strengthening and second-phase strengthening play important roles in Mg–Li alloys [17,18]. In addition to the primary second phase in the as-cast state, the dispersive reinforced phase dynamically precipitated during the deformation process also enables the optimization of the mechanical properties [19].

The present study compared the microstructures and mechanical properties at different temperature and deformation modes (R and MDR). High-density, dispersively distributed, nano-scale $MgZn_2$ particles and refined grains were found in the specimen deformed by cryogenic MDR. The relative strengthening mechanisms in the specimen are discussed.

## 2. Material and Methods

### 2.1. Preparation and Processing of Raw Materials

Mg–16Li–4Zn–1Er (wt.%) alloy was prepared in a vacuum medium-frequency induction furnace under the protection of Ar atmosphere. The ingots for the alloy were pure metals of Mg (>99.9 wt.%), Li (>99.9 wt.%), Zn (>99.9 wt.%), and a master alloy of Mg–20 wt.%Er. Then, the melt was poured into a permanent mold with dimensions of 120 mm × 110 mm × 40 mm. The 20 mm × 20 mm × 20 mm blocks for rolling were cut from the as-cast alloy. The rolling modes were R and MDR. The intended reduction between each pass was set at 0.8 mm. The specific steps of the processing are shown in Figure 1 (RD, rolling direction; TD, transverse direction; ND, normal direction). The total reduction of R was 60%, while thr A-side was always maintained as an RD–TD surface. The formula for the equivalent strain law of rolling process is as follows [20]:

$$\varepsilon = \frac{2}{\sqrt{3}} \left| \ln\left(\frac{h_0}{h}\right) \right|$$

In this experiment, the equivalent strain of MD rolling was stipulated to be in line with R, that is, $\varepsilon_{MDR} = \varepsilon_{R60\%} = 1.05804$. In MD rolling, A-side and B-side act as the RD–TD surface and successively consume half of the equivalent strain ($\varepsilon_{A-MDR} = \varepsilon_{B-MDR} = 1/2\varepsilon_{MDR}$). The details are shown in Figure 1. Importantly, after many experimental operations, it was shown that in the actual rolling process, due to the lateral flow of metal, when the thickness of sample was reduced from 20 mm to 12.6 mm, the width expanded from 20 mm to 23 mm.

For cryogenic rolling, the specimens were steeped in liquid nitrogen for 10 min before rolling. Similarly, they were placed back into liquid nitrogen for 5 min between each lane.

### 2.2. Microstructural Characterization

Transmission electron microscopy (TEM) and X-ray diffraction (XRD) were used for the analysis of the microstructures. The average grain size and second-phase size were determined by the mean linear intercept method.

### 2.3. Compression Test

The gleeble experiments were carried out at 25 °C and 100 °C. The size of the sample was φ8 mm × 12 mm. The position of the compression sample in the rolled samples is shown in Figure 1. The strain rate was $1.0 \times 10^{-2}$ s$^{-1}$, and the engineering strain was 70%. The mean value for each state was determined by the five compressed samples.

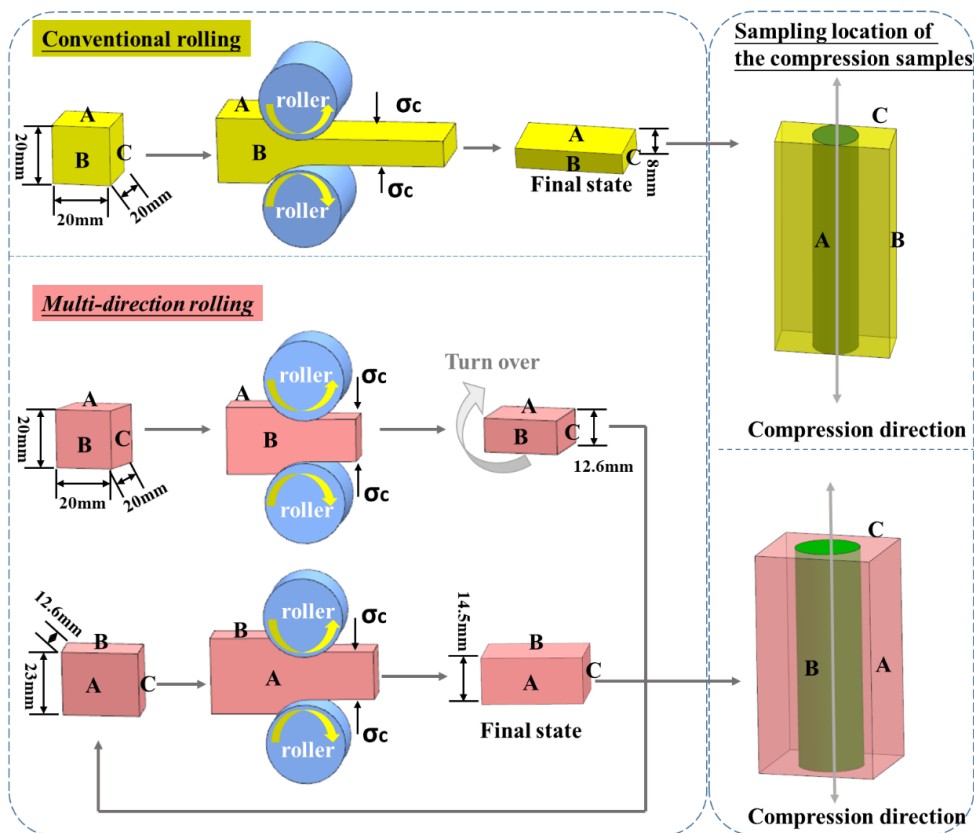

**Figure 1.** Schematic diagram of rolling and the positions of compression samples in as-rolled samples.

## 3. Results

### 3.1. Microstructures

The TEM bright-field images of the areas without the second-phase fragments were analyzed as shown in Figure 2. When the alloy grains were rolled in multiple directions at cryogenic temperature, a large range of uniformly distributed nano-grains appeared, with a size of about 56 nm. When a diffraction spot image was collected in any area of the nanocrystal, it contained all ring-shaped spots.

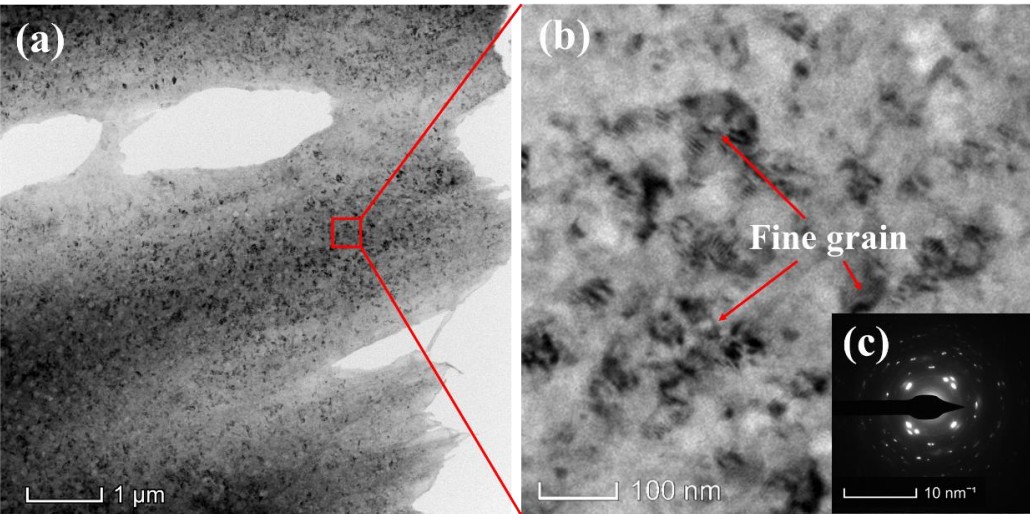

**Figure 2.** (**a**,**b**) Fine grains in bright-field TEM images of cryogenic MD-rolled alloy; (**c**) diffraction spot image.

It is worth noting that the nano-$MgZn_2$ phase was detected in the matrix of the cryogenic MD-rolled alloy. Figure 3a shows the bright-field TEM image area that was selected from the cryogenic MD rolled alloy for the measurement of diffraction spots. Figure 3b is a composite spot image of the β-Li and $MgZn_2$ phases. Figure 3c,d show dark-field TEM images of $MgZn_2$ under different crystal planes. In the LZE1641 alloy system, the primary second phase is a micron-sized second-phase particle, and there is no nano-sized second phase. Based on the results of our previous study [13], we inferred that this nanometer-sized densely distributed second phase was precipitated during the deformation process; that is, the morphology of small particles and diffraction spots in the dark field can be identified as the $MgZn_2$ precipitated phase. According to the present results, the precipitated phase was densely and uniformly distributed in the matrix with a size of about 35 nm. Figure 4 shows the XRD image of MD rolling at cryogenic temperature. In XRD, the peak of the $MgZn_2$ phase appeared on the (301) crystal plane.

The bright-field TEM images of the cross-section after compression at 100 °C are shown in Figure 5. After the hot compression, the grain size of the cryogenic rolled alloy was about 2~4 μm while the grain size of the cryogenic MD-rolled alloy was 1 μm. In addition, a large number of $MgZn_2$ particles still remained in the matrix. The TEM image of the nano-precipitated $MgZn_2$ phase area and the detection result of the element distribution are shown in Figure 6. We found that the average size of the nano-phase $MgZn_2$ phase was about 40 nm, which is the same as the size before compression. The distribution of Zn was consistent with the distribution of the nano-precipitated phase. This demonstrates that the $MgZn_2$ phase could not be fragmented after hot compression and is conducive to hindering the migration of grain boundaries during said compression [21].

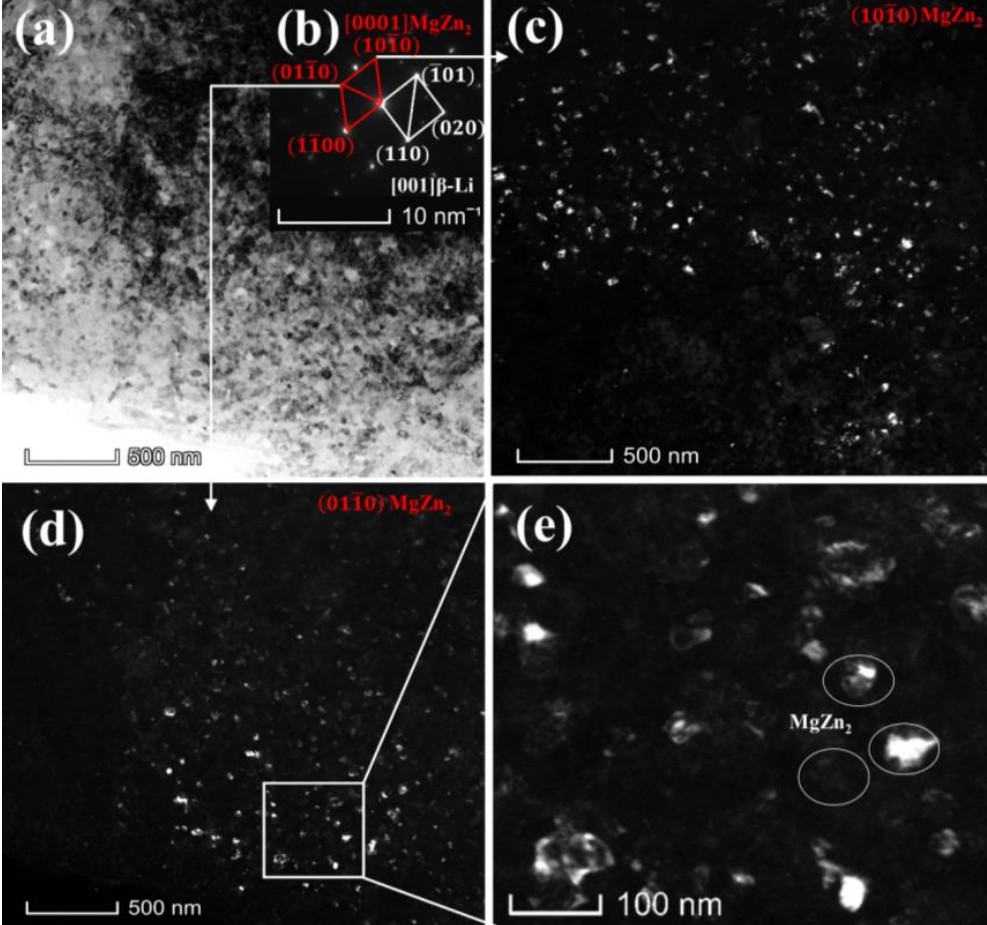

**Figure 3.** (**a**) Bright-field TEM images of cryogenic MD-rolled alloy; (**b**) diffraction spot image of (**a**); (**c**–**e**) dark-field TEM images of nano-$MgZn_2$ phases.

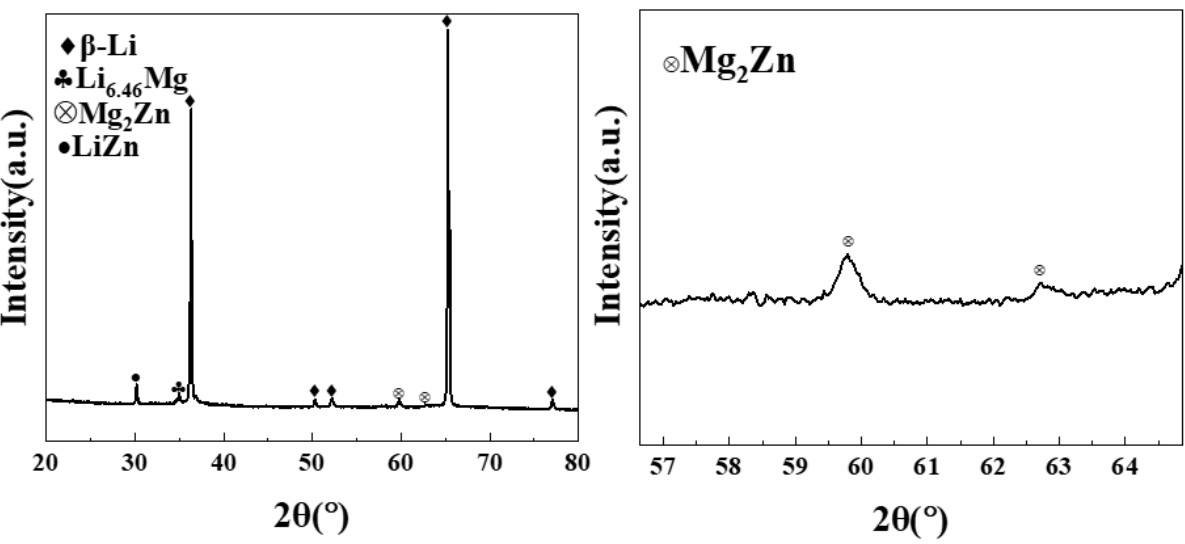

**Figure 4.** XRD images of cryogenic MD-rolled alloy.

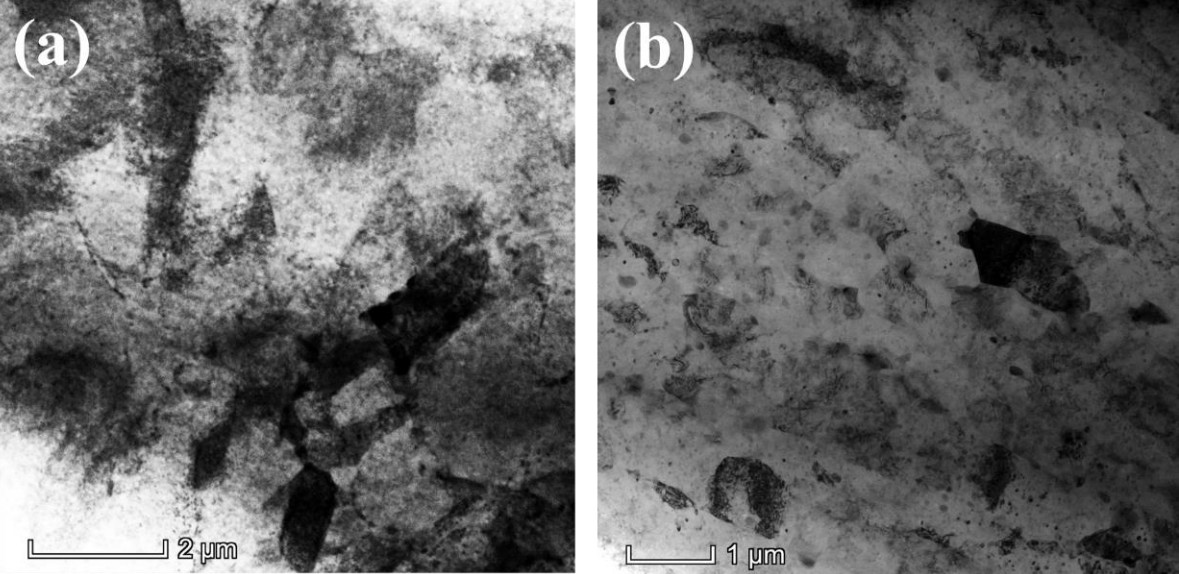

**Figure 5.** Bright-field TEM images of cross-section morphologies after 100 °C compression, (**a**) cryogenic conventionally rolled alloy; (**b**) cryogenic MD-rolled alloy.

As shown in Figure 7, while the dislocations are arranged in a disordered manner at the grain boundary, the nano-precipitation particles further aggravated the proliferation and packing of the dislocations. Multiple dislocation lines are intertwined and entangled with each other, providing an effect for the work-hardening of the alloy. Based on these results, $MgZn_2$ only existed in the MD-rolled alloy at the cryogenic temperature. This indicates that $MgZn_2$ is a stress-induced nano-precipitation phase. The type, structure, and shape of the second phase depend on the elastic strain energy inside the alloy [22]. To eliminate the concentrated stress level of the cryogenic MD-rolled alloy, spherical nano-$MgZn_2$ with hcp structure is precipitated. This nano-precipitated phase is difficult for dislocations to cut and can effectively pin their movement. This promotes the intertwining phenomenon of dislocations [23]. In the cryogenic environment, dislocations cannot be released immediately, which further enhances the number of dislocations and the degree of dislocation packing in the matrix. Thus, dislocation strengthening was further optimized.

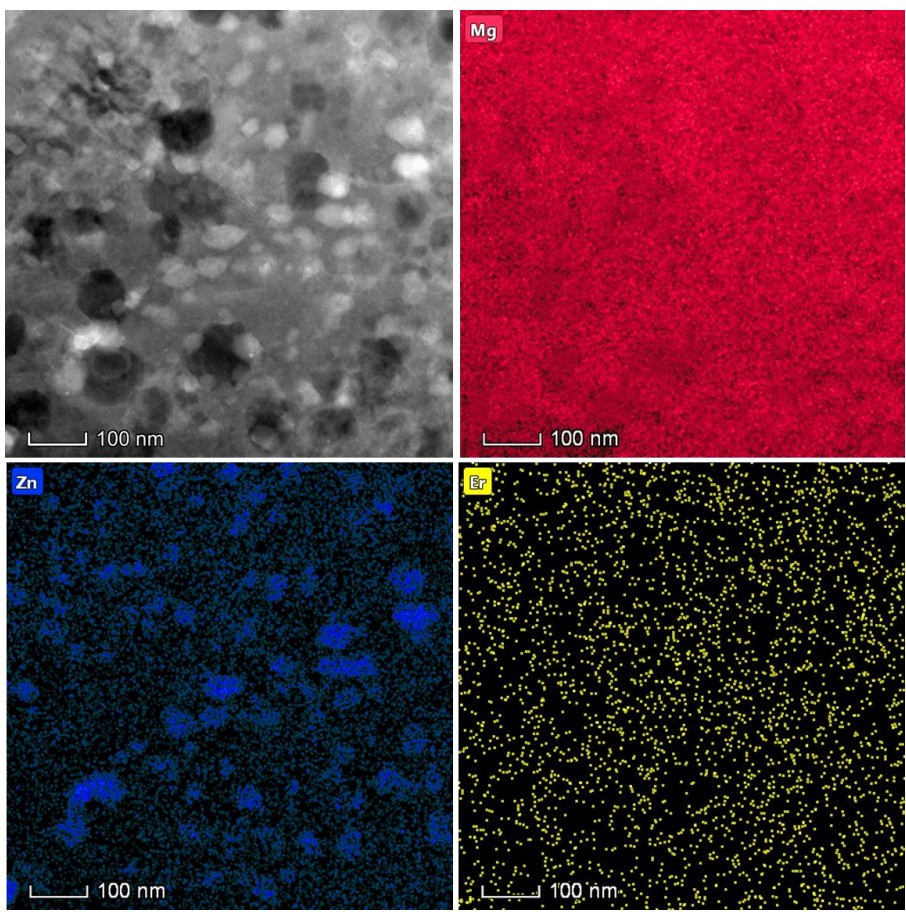

**Figure 6.** Mappings of nano-MgZn$_2$ phases in cryogenic MD-rolled alloy after 100 °C compression.

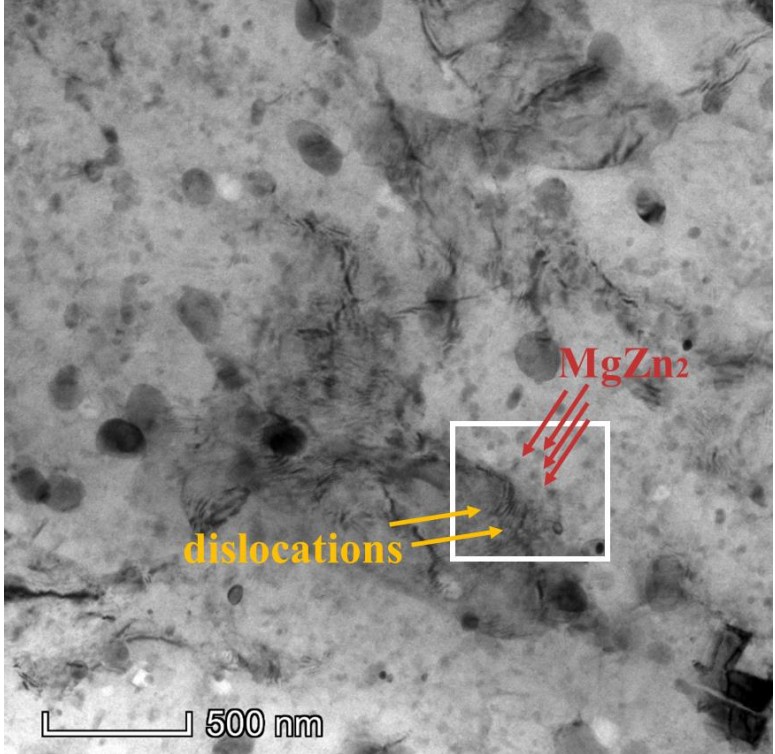

**Figure 7.** Bright-field TEM images for the interaction between nano-MgZn$_2$ phases and dislocations.

### 3.2. Mechanical Properties

The true stress–strain curves of cryogenic rolled alloys compressed at 25 °C (room temperature) and 100 °C are shown as Figure 8. The compressive strength of the MDR alloy was 331 MPa, which was 21% higher than that of the conventional rolled alloy (273 MPa). After compression at 100 °C, the compressive strength of the conventional rolled alloy was 192 MPa, while the MDR alloy could still maintain its compressive strength of 238 MPa.

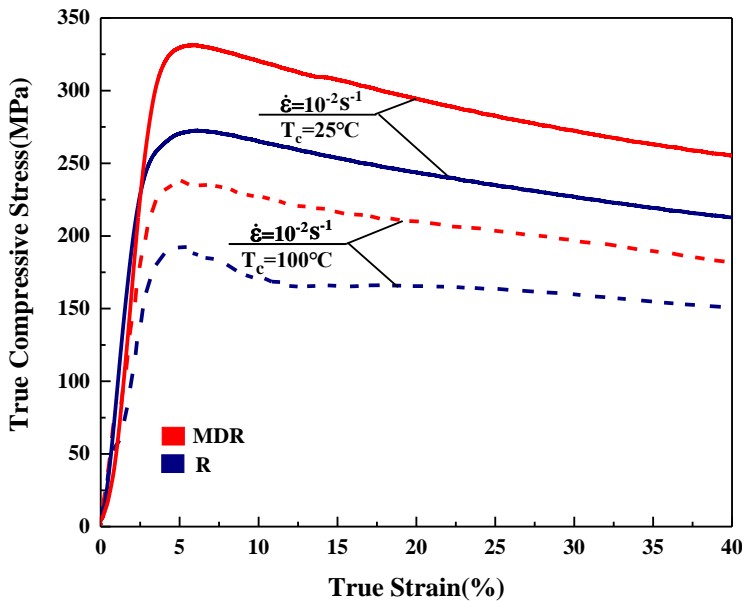

**Figure 8.** Compressive true stress–true strain curves of cryogenic rolled alloys.

### 4. Discussion

### 4.1. Precipitation of the MgZn$_2$ Phase

In our previous experimental results, we proved that low temperature deformation accumulates more stress concentration and deformation energy storage in the matrix compared with the same process at room or elevated temperature [13]. This energy storage provides a huge impetus for DRX nucleation and atomic diffusion. The phase precipitation and growth are essentially behaviors of the polarization and growth during atomic diffusion [24,25]. Deformation can form a large number of defects, such as dislocations. They can provide more nucleation sites for second-phase precipitation, and can also become fast diffusion channels for the solute atoms and accelerate the atomic diffusion.

According to the results, the sequence of precipitation for the Mg–Zn phase was $\beta_1'(Mg_4Zn_7) \rightarrow \beta_2'(MgZn_2) \rightarrow \beta(MgZn)$ [25,26]. In this experiment, due to the limitation of the detection means, it cannot be determined whether there was a Mg$_4$Zn$_7$ or MgZn mixture, and only MgZn$_2$ was detected. Thus, we concluded that MD rolling mainly led to the precipitation of the MgZn$_2$ phase. Under the stress input provided by rolling, solute atoms still have a certain diffusion capacity, so an oversaturated matrix still has the ability to produce a second-phase precipitation. However, the low-temperature environment is not conducive to this reaction. That is, low temperature inhibits the precipitation of an equilibrium precipitation phase [27]. The balance between high energy storage and low temperature conditions causes an incomplete reaction. This prompts the emergence of the non-equilibrium precipitate product. This is why MgZn$_2$ exists as a non-primary phase in the matrix of MD-rolled alloy at cryogenic temperature.

However, it is noteworthy that the presence of the MgZn$_2$ phase was not detected in conventional low-temperature rolling. The type, structure, and shape of the second phase depends on the strain energy inside the alloy [22]. The shear stress in the XY direction inside the alloy during stress loading is shown in Figures 9 and 10. The difference in the shear stress inside the alloy also reflects the difference in the diffusion law of the solute atoms in the alloy [28]. Compared with conventional rolling, there is a stress-loading

axis transformation in the MDR process, which increases the crystal defects inside the alloy (dislocations, vacancies, etc.). Such a change raises the energy of the alloy, and the whole system reaches an unstable state. To stabilize the system, a new phase forms cores at the defect positions to release energy, thus reducing the free energy of the system. In other words, MDR makes $MgZn_2$ phase precipitation more prone. The critical radius of nucleation for the precipitated phase is the following [29]:

$$r^* = \frac{2\gamma}{\Delta G_v + \Delta G_\varepsilon + \Delta G_D},$$

where $\gamma$ is the interface energy between the new phase and the parent phase; $\Delta G_v$ is the free energy difference between new phase and the matrix in the unit volume; $\Delta G_\varepsilon$ is the strain energy of the new phase per unit volume; and $\Delta G_D$ is the reduced-system free energy caused by the nucleation of $MgZn_2$ on the defect. Therefore, the increasing of the defect increases $\Delta G_D$, decreases the radius of the critical nucleation, and the number of nuclei increases, so that the driving force of the nuclei increases. This results in the fine-grained size of the precipitated phase. A large number of $MgZn_2$ phases triggers a huge total surface energy. The spherical particles have the lowest surface energy in all shapes, so the precipitated phase gradually tends to globularity after nucleation. Thus, to eliminate the concentrated stress level, a spherical (2H) Laves nano-$MgZn_2$ phase with hcp structure was precipitated in the cryogenic MD-rolled alloy [30]. The coupling of stress direction to crystal orientation reduces the lattice mismatch and solute diffusion rate. In the process of uniform isothermal precipitation, the driving force of the precipitation $\Delta G$ can be described by the following formula [31]:

$$\Delta G = -\frac{kT}{V_{at}} \ln\left(\frac{C}{C_{eq}}\right)$$

where $V_{at}$ is the solute atomic volume; $C_{eq}$ is the solubility of $MgZn_2$ in the alloy at equilibrium; C is the current solubility of $MgZn_2$ in the alloy; $k$ is the Bozmann constant; and $T$ is the absolute temperature. Therefore, the lower the $T$, the smaller the precipitation driving force of the second phase. The second-phase nucleus does not necessarily grow up directly, but only grows up when the size exceeds a certain critical value. At low temperatures, the driving force is insufficient to sustain the $MgZn_2$ phase size. The explosive uniform nuclear barrier and the tendency to coarsening of the precipitated phase are suppressed. Therefore, we ultimately obtained a high-density and uniformly distributed $MgZn_2$ nanoparticle precipitate phase in the cryogenic MDR alloy.

*4.2. The Strengthening Effect of MgZn2*

$MgZn_2$ hinders the migration of grain boundaries [32,33]. The velocity $v$ of the grain boundary when passing through a single particle obeys the following formula [34]:

$$v = \left(\frac{L}{\sigma_{gb}}\right)\left(\Delta f - \Delta f_{drag}\right)$$

where $L$ is the mobility; $\sigma_{gb}$ is the grain boundary energy; $\Delta f$ is driving force; $\Delta f_{drag}$ is the dragging force. Obviously, in the process of grain boundary migration, the contact area between the precipitated phase and the grain boundary becomes larger and larger. The drag force of particles on grain boundary migration increases, and the speed of grain boundary migration decreases accordingly. When there are multiple second-phase particles, the pinning force $F_A$ received by the grain boundary follows the following formula [34]:

$$F_A = \sigma_{gb} sin\theta'\left(\frac{2}{d - 2rcos\theta} - 1\right)$$

The significance of each parameter is indicated in Figure 11. Both the large number of precipitated phases and the fine particle spacing can effectively enhance the hindrance of grain boundary migration. Thus, nano-particles can reduce the peak of the function between grain-size distribution and time. This implies that $MgZn_2$ can effectively impede the disappearance of small-size grains [35]. Therefore, the cryogenic MD-rolled alloy possesses the smallest grains, and the grains still maintain the finest size among the six states after high temperature compression.

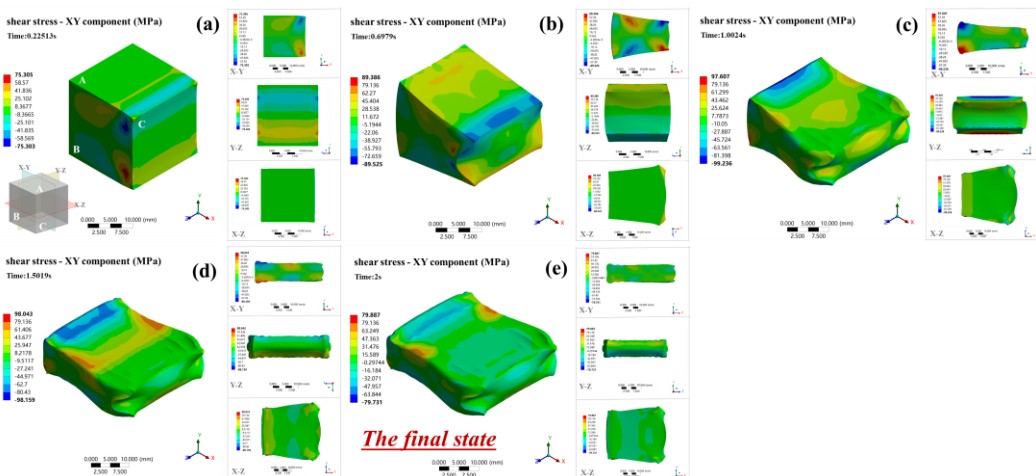

**Figure 9.** The shear–stress distribution of conventionally rolled alloys. The graphs are cut from different time points of deformation. (**a**) 0.22513 s; (**b**) 0.6979 s; (**c**) 1.0024 s; (**d**) 1.5019 s; (**e**) 2 s.

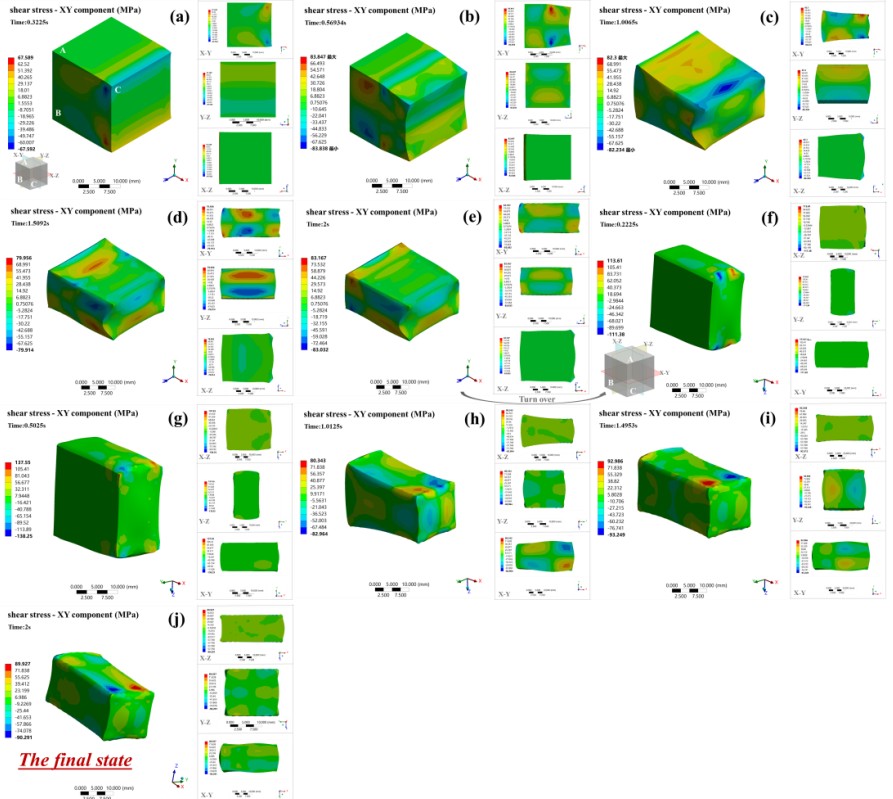

**Figure 10.** The shear–stress distribution of MD–rolled alloys. The graphs are cut from different time points of deformation. (**f**–**j**) are redefined according to the time after the specimen is turned over. (**a**) 0.3225 s; (**b**) 0.56934 s; (**c**) 1.0065 s; (**d**) 1.5092 s; (**e**) 2 s; (**f**) 0.2225 s; (**g**) 0.5025 s; (**h**) 1.0125 s; (**i**) 1.4953 s; (**j**) 2 s.

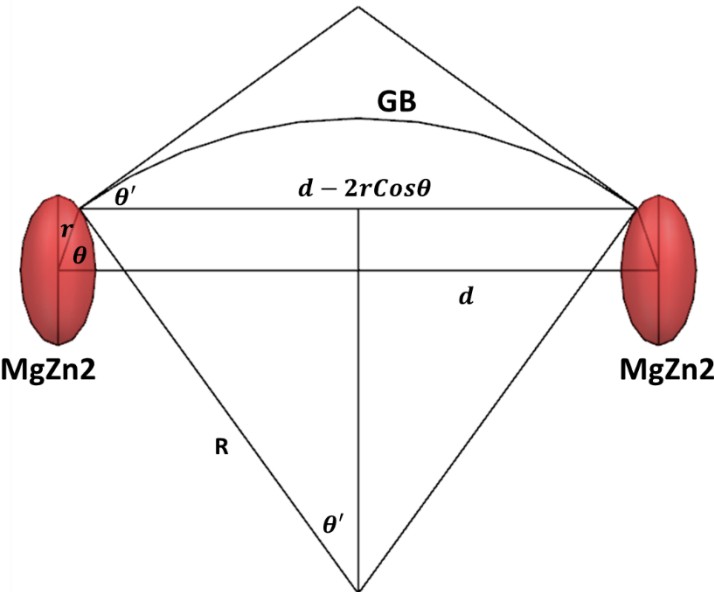

**Figure 11.** Dragging of the grain boundary migration by two MgZn$_2$ particles.

　　This nano-precipitated phase is difficult for dislocations to cut and can effectively pin their movement. This promotes the intertwining phenomenon of dislocations [23]. In the cryogenic environment, dislocations cannot be released immediately, which further enhances the number of dislocations and the degree of dislocation packing in the matrix. Thus, dislocation strengthening is further optimized.

**5. Conclusions**

1. Cryogenic MDR triggers the formation of a large number of nano-grains.
2. The differences in MDR and R deformation lead to shear-stress changes inside the alloy.
3. A large amount of uniformly dispersed nano-precipitation-phase MgZn$_2$ appears only in the cryogenic MDR LZ1641 alloy.
4. MgZn$_2$ has an obstructive effect on the migration of grain boundaries.
5. MgZn$_2$ cannot be cut by dislocations, by which the effect of dislocation strengthening is consolidated.

**Author Contributions:** Q.J.: Study design, data analysis, data interpretation, writing; X.M.: literature search, data analysis; R.W.: data analysis; S.J.: figures; J.Z.: data collection; L.H.: literature search. All authors have read and agreed to the published version of the manuscript.

**Funding:** This paper was supported by Ph.D. Student Research and Innovation Fund of the Fundamental Research Funds for the Central Universities (3072020GIP1015), National Natural Science Foundation of China (51871068, 51971071, 52011530025, U21A2049, 52271098), Fundamental Research Funds for the Central Universities (3072022QBZ1002), National Key Research and Development Program of China (2021YFE0103200).

**Data Availability Statement:** Not applicable.

**Conflicts of Interest:** The authors declare no conflict of interest.

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
