# Peer review of "BCC-Based Mg–Li Alloy with Nano-Precipitated MgZn2 Phase Prepared by Multidirectional Cryogenic Rolling"

_metals, doi:10.3390/met12122114_

Round 1

Reviewer 1 Report

1. The SAED pattern shown in Figure 3 should be solved. I see there diffraction spots of two phases. Probably magnesium and MgZn2.

2. From which diffraction spot was the dark field image taken?

3. How do the authors distinguish the MgZn2 precipitates in Fig. 3?

4. Do the authors have a higher magnification TEM image of MgZn2 precipitates?

5. The XRD pattern is not readable. Please increase font size.

6. Page 5, line 125 - The average size of MgZn2 nanophase is about 40 nm, which is the size before compression? Where are the particle size analyses of the sample before compression?

Reviewer 2 Report

1. It is not clear why the authors indicated the mechanical treatment as a process at minus 196 C. The samples were cooled down to this temperature, thus, it can be named by this way, but it is not real temperature of the process according to given description.

2. According to given description after MD rolling the grain size is around 54 nm and particle size 35 (section 3.1 and fig.3). After compression at 100 C the grain size increased significantly, while the size of the particles is almost the same. The state after compression is confirmed perfectly by mapping and TEM images (Fig.6,7), while the picture of the samples after MD rolling does not allowed to determine these parameters. Is it possible to demonstrate in order to have a picture with the same magnification or for rolled samples or it is not possible due to their deformed structure?

3. It does not clear why the authors does not discuss the properties after compression at 100 C?

4. The equation for pinning force (section 4.2) must be clarified or (if it is already known) the Reference must be indicated.

Reviewer 3 Report

The article is very interesting and the results are well presented. 

I would like to suggest some small changes in TEM investigations presentations.  In Figures 1 and 2 diffraction patterns are described as FFT patterns. FFT patterns are used for HRTEM observations.  

Furthermore, I would suggest large magnifications for the observation of fine grains and Dark Field imaging.  It will show the magnitude of refinement.  Also, the Dark Filed image in Fig. 2 should be in higher magnifications.
